# Detection of *Vibrio parahaemolyticus* Based on Magnetic and Upconversion Nanoparticles Combined with Aptamers

**DOI:** 10.3390/foods12244433

**Published:** 2023-12-11

**Authors:** Xinjie Song, Wei Li, Li Wu, Tianfeng Lv, Yao Zhang, Juan Sun, Xuping Shentu, Xiaoping Yu, Yuanfeng Wu

**Affiliations:** 1Zhejiang Provincial Key Laboratory of Chemical and Biological Processing Technology for Agricultural Products, School of Biological and Chemical Engineering, Zhejiang University of Science and Technology, Liuxia Street Number 318, Hangzhou 310023, China; xjsong@zust.edu.cn (X.S.); wuli@zust.edu.cn (L.W.); LTF107@163.com (T.L.); zhangyao@zust.edu.cn (Y.Z.); sunjuan18@zust.edu.cn (J.S.); 2Korean Medicine (KM) Application Center, Korea Institute of Oriental Medicine, Daegu 41062, Republic of Korea; liwei1986@kiom.re.kr; 3Zhejiang Provincial Key Laboratory of Biometrology and Inspection and Quarantine, College of Life Science, China Jiliang University, Hangzhou 314423, China; stxp@cjlu.edu.cn (X.S.); yxp@cjlu.edu.cn (X.Y.)

**Keywords:** *Vibrio parahaemolyticus*, aptamer, magnetic nanoparticles, upconversion nanoparticles, detection

## Abstract

*Vibrio parahaemolyticus* is a halophilic and heat-labile gram-negative bacterium and is the most prevalent foodborne bacterium in seafood. In order to develop a rapid and sensitive method for detecting the foodborne pathogenic bacterium *Vibrio parahaemolyticus*, an aptamer-modified magnetic nanoparticle and an aptamer-modified upconversion nanoparticle were synthesised and used as a capture probe and a signal probe, respectively. The aptamer-modified magnetic nanoparticle, *V. parahaemolyticus* cell, and aptamer-modified upconversion nanoparticle formed a sandwich-like complex, which was rapidly separated from a complex matrix using a magnetic force, and the bacterial concentration was determined by fluorescence intensity analysis. The results showed that the fluorescence intensity signal correlated positively with the concentration of *V. parahaemolyticus* in the range of 3.2 × 10^2^ to 3.2 × 10^5^ CFU/mL, with a linear equation of y = 296.40x − 217.67 and a correlation coefficient of R^2^ = 0.9610. The detection limit of the developed method was 4.4 CFU/mL. There was no cross-reactivity with other tested foodborne pathogens. This method is highly specific and sensitive for the detection of *V. parahaemolyticus*, and can achieve the qualitative detection of this bacterium in a complex matrix.

## 1. Introduction

Foodborne illnesses caused by pathogenic microorganisms are common and result in many food safety challenges [1]. Seafood is a nutrient-rich food source, and its consumption has been increasing worldwide in recent decades. The main safety concern surrounding this increase in consumption is foodborne pathogen-induced infections [2]. Common pathogenic bacteria in seafood include *Vibrio*, *Salmonella*, *Listeria*, *Shigella*, *Staphylococcus*, *Clostridium*, and *Escherichia coli* [3,4]. *Vibrio parahaemolyticus*, a halophilic and heat-labile gram-negative bacterium, is the most prevalent foodborne bacterium in seafood. *V. parahaemolyticus* contamination can lead to gastroenteritis, infection, and sepsis [5]. *V. parahaemolyticus*-caused diseases have been frequently reported all over the world in recent years, making *V. parahaemolyticus* a major concern in seafood safety. A study showed that 4256 patients were infected by *V. parahaemolyticus* between 2003 and 2016 in Korea [6]. Chen et al. reported that, among 410 outbreaks of bacterial foodborne illnesses occurring from 2015 to 2020 in Zhejiang province, China, 56.69% (232 cases) were caused by *V. parahaemolyticus* [7]. Another report mentioned that vibriosis was responsible for approximately 80,000 illnesses and 100 deaths annually in the United States [8]. Recently, *Vibrio* spp. infections have attracted more concern because climate warming and global trade increases might increase human *Vibrio* infections worldwide [9]. Therefore, rapid and reliable *V. parahaemolyticus* detection is essential in the control of food safety.

Current *V. parahaemolyticus* detection methods include conventional microbiological culturing [10], molecular biological detection [11], and immunological detection [12]. Hu et al. [13] reported a nanogold-assisted HRM-qPCR method for detecting *V. parahaemolyticus*, in combination with self-screening and traditional targeting, with a detection limit of 5 × 10^1^ CFU/mL. Feng et al. [14] developed a recombinase-aided amplification assay for detecting *V. parahaemolyticus.* This assay could detect concentrations of 7 × 10^3^ CFU/mL directly and could detect levels as low as 0.1 CFU/mL when a 4 h enrichment was performed. Zeng et al. [15] reported the development of a PCR-based lateral flow test strip for detecting *V. parahaemolyticus* in codfish, with a detection limit of 50 CFU/mL. Similarly, Yang [16] developed a fluorescence immunoassay based on the inner filter effect and BSA-gold nanoclusters. Each of these methods has advantages and disadvantages, and developing new methods for the rapid and sensitive detection of *V. parahaemolyticus* is crucial for the surveillance of foodborne diseases.

Upconversion nanoparticles (UCNPs) are novel types of fluorescent materials that exhibit a special luminescence phenomenon: the wavelength of the emitted light is less than the wavelength of the excited light, that is, upconversion luminescence [17]. In the 1960s, Bloembergen first proposed the phenomenon of upconversion, and Porter proved that the upconversion process is a nonlinear optical process that converts two or more low-energy photons into high-energy photons [18]. With the development of research, rare-earth-doped upconversion nanoparticles have been found to have remarkable optical properties, including high resistance to light bleaching, a sharp emission bandwidth, a long lifetime, excellent spectral characteristics, and high photostability. Upconversion luminescence nanoparticles can emit visible or ultraviolet light upon excitation with near-infrared light and are widely applied in biosensing for food safety. Compared to conventional organic fluorescent dyes and quantum dots, the unique anti-Stokes shift luminescence signal of UCNPs can be used to identify autofluorescence in complex real matrices, such as in food, the environment, and biological tissues, thereby reducing background interference and improving sensitivity. Aptamers are synthetic short single-stranded oligonucleotides (DNA or RNA) screened from random DNA libraries in vitro by SELEX [19]. Aptamers have attracted much attention in the field of food safety detection due to their affinity and specificity for targets [20]. Detection methods based on aptamers were studied for the analysis of foodborne pathogens including *V. parahaemolyticus* [21]. Furthermore, magnetic immobilisation separation using Fe_3_O_4_ magnetic nanoparticles as carriers to enrich pathogens in microbiology can enhance the analysis selectivity and sensitivity [22]. Therefore, in this study, we developed a novel *V. parahaemolyticus* detection method using aptamer-based magnetic fluorescent nanoprobes. Magnetic and upconversion nanoparticles were combined with amino-modified aptamers to capture target pathogenic bacteria based on aptamer specificity. *V. parahaemolyticus* levels were quantitatively analysed based on the fluorescence intensity.

## 2. Materials and Methods

### 2.1. Materials

FeCl_3_·6H_2_O and 1,6-hexanediamine were purchased from Shanghai Macklin Biochemical Co., Ltd. (Shanghai, China). Ethylene glycol, ethanol, methanol, isopropanol, and sodium hydroxide were purchased from Shanghai Lingfeng Chemical Reagent Co., Ltd. (Shanghai, China), and 25% glutaraldehyde was purchased from Yonghua Chemical Co., Ltd. (Shanghai, China). YCl_3_, YbCl_3_, ErCl_3_, 1-octadecene (ODE), and ammonium fluoride (NH_4_F) were purchased from Aladdin Reagent Co., Ltd. (Shanghai, China). The 3-aminopropyltriethoxysilane (APTES), 0.01 mol/L phosphate-buffered saline (PBS, pH 7.4), oleic acid (OA), sodium acetate, tetraethyl orthosilicate (TEOS), and *V. parahaemolyticus* aptamer (5′-NH_2_-TCTAAAAAATGGGCAAAAAAACATGACTCGTTGAGATACT-3′) used herein were purchased from the Shanghai Sangon Biological Science & Technology Company. Alkaline peptone water medium and thiosulfate citrate bile salt sucrose agar medium were purchased from Beijing Landbridge Technology Co., Ltd. (Beijing, China). *V. parahaemolyticus* (CGMCC 1.1997), *Escherichia coli* (CGMCC 1.12883), *Staphylococcus aureus* (CGMCC 1.6750), and *Salmonella* (CGMCC 1.10603) were purchased from the China General Microbial Strain Preservation and Management Centre. Enterotoxigenic *Escherichia coli* (ATCC 35401) was purchased from the American Type Culture Collection.

### 2.2. Apparatus

A JEM 2100F high-resolution field emission transmission electron microscope (TEM, JEOL, Tokyo, Japan), X-ray diffractometer (Malvern Panalytical, Malvern, UK), high-resolution transmission electron microscope (TEM, HITACHI H-7600; HITACHI, Tokyo, Japan), fluorescence spectrophotometer (F-4500 HITACHI, Tokyo, Japan), and Fourier transform infrared spectrometer (FTIR, BRKER, Saarbrucken, Germany) were used.

### 2.3. Synthesis of Amine-Functionalised Fe_3_O_4_ Magnetic Nanoparticles

Amine-functionalised Fe_3_O_4_ magnetic nanoparticles were prepared as previously described [23,24]. Briefly, anhydrous sodium acetate (2.0 g), 1,6-hexanediamine (6.5 g), and FeCl_3_·6H_2_O (1.0 g) in glycol (30.0 mL) were mixed at 50 °C. The mixture was allowed to react for 10 h at 200 °C. After being cooled to room temperature, the products were separated by a magnetic stirrer before being collected, washed with ultrapure water (3 times) and ethanol (3 times), and dried at 60 °C for 8 h. After drying, a black powder consisting of amino-functionalised Fe_3_O_4_ magnetic nanoparticles was collected. TEM imaging, XRD patterns, FTIR spectra, and VSM magnetisation curves were used for the property analysis of the Fe_3_O_4_ magnetic nanoparticles.

### 2.4. UCNP Synthesis and Surface Modifications

To synthesise UNCP, firstly, 1 mmol of ReCl_3_ (Y:Yb:Er = 78:20:2), 6 mL of OA, and 15 mL of ODE were concurrently added to a 100 mL three-necked flask; the mixed solution was heated for 30 min at 160 °C and allowed to cool to 50 °C [25]. Subsequently, 10 mL of a methanol solution, containing 2.5 mmol of NaOH and 3.9 mmol of NH_4_F, was added to the mixed solution and agitated for 30 min. Afterwards, the mixture was heated for 1 h at 100 °C and then 300 °C under the protection of argon gas, and kept at the latter temperature for 30 min. After cooling to room temperature, the product was precipitated using ethanol and was washed and centrifuged thrice. The final product was dried for 12 h at 60 °C [26].

Surface modification of the UCNPs was performed as previously described [27]. Firstly, 20 mg of the synthesised UCNPs was ultrasonicated for 30 min in 60 mL of isopropanol. Subsequently, 20 mL of distilled water and 2.5 mL of 25% ammonia were rapidly added to the mixture, and it was magnetically stirred. Afterwards, 20 mL of isopropanol and 60 µL of tetraethoxysilane (TEOS) were added to the solution and allowed to react for 3 h, before 30 mL of isopropanol and 200 µL of APTES were added. The product was left to stand for 2 h at room temperature after 1 h of reaction, before being washed and centrifuged several times. Finally, the product was vacuum-dried for 12 h at 60 °C. The amino-functionalised UCNPs were collected and kept at 4 °C until subsequent use. The dried powders of UCNPs and UCNPs@SiO_2_ were used to perform the characterisation of TEM imaging, XRD patterns, and FTIR spectra. To prevent the influence of aggregation on the result, both the UCNPs and UCNPs@SiO_2_ nanoparticles were uniformly suspended in water, and the phase and fluorescence spectra were measured with excitation and emission spectra of 980 nm and in the 400–800 nm region directly.

### 2.5. Synthesis of Aptamer-Fe_3_O_4_ Magnetic Nanoparticles and Aptamer-UCNPs

Amino-functionalised Fe_3_O_4_ magnetic nanoparticles (2 mg) were ultrasonicated in 1 mL of PBS (pH 7.4) for 30 min. Subsequently, 250 µL of 25% glutaraldehyde was added to the mixture, and the solution was shaken for 2 h at 25 °C. The product was magnetically separated and washed three times with PBS and then dispersed in 1 mL of PBS. Afterwards, 20 µL of the 25 μM aptamer was added to the uniform amino-functionalised Fe_3_O_4_ magnetic nanoparticle solution, and the mixture was shaken for 12 h at 37 °C, magnetically separated, and washed thrice with PBS. The final product was dispersed in 1 mL PBS and stored at 4 °C [28]. The aptamer in the supernatant before and after the reaction was measured using UV-vis spectroscopy to confirm the immobilisation of the aptamer on the Fe_3_O_4_ magnetic nanoparticles.

The aptamer-UCNP preparation was similar to the preparation of the aptamer-Fe_3_O_4_ magnetic nanoparticles. First, the amino-functionalised UCNPs were ultrasonicated in 1 mL of PBS (pH 7.4) for 30 min. Subsequently, 250 µL of 25% glutaraldehyde was added, and the mixture was shaken for 2 h at 25 °C. The product was separated by centrifugation (12,000 rpm, 10 min) and washed thrice with PBS before being dispersed in 1 mL of PBS. Afterwards, 40 µL of the 25 µM aptamer was added to the uniform amino-functionalised UCNP solution, and the mixture was shaken for 12 h at 37 °C. The mixture was separated via centrifugation and washed four times with PBS. The final product was dispersed in 1 mL of PBS and stored at 4 °C. The aptamers in the supernatant before and after the reaction were measured using UV-vis spectroscopy to confirm the immobilisation of the aptamer on the UNCPs. Different concentrations of UNCPs (1, 2, 4, and 6 mg) were tested to optimise the synthesis of the aptamer-UCNPs.

### 2.6. Optimization of Reacted Amount of Aptamer-Fe_3_O_4_ Magnetic Nanoparticles and Aptamer-UCNPs

To optimise the amount of reacting aptamer-Fe_3_O_4_ magnetic nanoparticles, 40, 80, 120, 160, and 200 µL of aptamer-MNPs were added to a fixed amount (150 µL) of aptamer-UCNP solution. Secondly, 100 µL of *V. parahaemolyticus* (at 10^5^ CFU/mL) was added to the aptamer-UCNP solutions simultaneously, and the mixture was replenished to 500 µL with PBS buffer and shaken slowly for 2 h at 37 °C. After reacting, the product was magnetically separated and washed thrice with PBS to remove unbound components. Finally, the product was resuspended in 400 µL of PBS buffer, and the fluorescence intensity of the suspension was measured at 542 nm using a fluorescence spectrophotometer with an excitation spectrum of 980 nm and emission spectra in the 400–800 nm region.

Afterwards, to optimise the amount of reacting aptamer-UCNPs, 50, 100, 150, 200, and 250 µL of aptamer-UCNPs (at 6 mg/mL) were added and reacted with the aptamer-Fe_3_O_4_ magnetic nanoparticles following the aforementioned procedure. The optimal amount of aptamer-UCNPs was determined based on the fluorescence intensity of the suspension, which was measured at 542 nm using a fluorescence spectrophotometer with an excitation spectrum of 980 nm and emission spectra in the 400–800 nm region.

### 2.7. Detection of V. parahaemolyticus Using the Developed Fluorescent Method

To detect *V. parahaemolyticus* using our developed method, 100 µL of *V. parahaemolyticus* was added to 500 µL of the aptamer-Fe_3_O_4_ magnetic nanoparticles suspension and gently vibrated for 2 h at 37 °C. The reactant was magnetically separated and washed thrice with PBS buffer. The aptamer-UCNP suspension was then added, and the mixture was gently vibrated for 1 h at 37 °C. Subsequently, the product was magnetically separated, washed thrice with PBS buffer, and resuspended in 400 µL of PBS buffer. The fluorescence intensity of the suspension was measured in the 400–800 nm region using a fluorescence spectrophotometer with an excitation spectrum of 980 nm. *V. parahaemolyticus* concentrations from 3.2 × 10^2^ CFU/mL to 3.2 × 10^5^ CFU/mL were tested to determine the detection limit of our method.

### 2.8. Detection Specificity

Four foodborne pathogenic bacteria, *Escherichia coli*, *Enterotoxigenic Escherichia coli*, *Salmonella*, and *Staphylococcus aureus*, were used to evaluate the specificity of the developed method. The tested foodborne pathogenic bacteria at a concentration of 10^5^ CFU/mL were mixed with the aptamer-Fe_3_O_4_ magnetic nanoparticles, reacted under gentle shaking, and separated using a magnetic force. Next, the separated mixture was mixed with aptamer-UNCPs and separated using a magnetic force after 30 min of reaction under gentle shaking. The separation was uniformly resuspended in PBS buffer by ultrasonication. The fluorescence intensities of the suspensions were measured using excitation and emission spectra of 980 and in the 400–800 nm region, respectively.

## 3. Results and Discussion

### 3.1. Mechanism of V. parahaemolyticus Detection

In our study, a *V. parahaemolyticus* detection method was developed using aminated nanoparticles and amino-modified aptamers. As the aptamer specifically binds to the target, aptamer-modified nanoparticles can identify and capture *V. parahaemolyticus*. Aptamer UCNPs and aptamer-Fe_3_O_4_ magnetic nanoparticles were used as fluorescent signals and a separation medium, respectively, in this study. First, the tested sample was mixed with aptamer-Fe_3_O_4_ magnetic nanoparticles and allowed to react under gentle shaking for 30 min. Aptamer-Fe_3_O_4_ magnetic nanoparticles could capture *V. parahaemolyticus* using aptamers when *V. parahaemolyticus* was present in the tested sample. The aptamer-Fe_3_O_4_-*V. parahaemolyticus* can be separated using a magnetic force. The separated aptamer-Fe_3_O_4_-*V. parahaemolyticus* was mixed with aptamer-UNCPs to create sandwich-like aggregates, aptamer-Fe_3_O_4_-*V. parahaemolyticus*-UNCPs, which can be formed and separated using a magnetic force (Figure 1). Due to the presence of UCNPs, the fluorescence intensity of the sandwich-like aggregate can be measured via fluorescence spectrophotometry with an excitation spectrum of 980 nm and emission spectra in the 400–800 nm region, and the intensity of this signal can be increased by increasing the concentration of *V. parahaemolyticus* for detection.

### 3.2. Characterisation of Amine-Functionalised Magnetic Fe_3_O_4_ Nanoparticle

Bare Fe_3_O_4_ magnetic nanoparticles must undergo surface modifications because of their high activity levels and susceptibility to oxidation. The TEM images (Figure 2A) indicated that the Fe_3_O_4_ magnetic nanoparticles were uniformly spherical with a size of 50 nm. The XRD results (Figure 2B) indicated that the Fe_3_O_4_ magnetic nanoparticles have typical X-ray diffraction lines at 2θ = 18.6°, 35.8°, 43.7°, 53.9°, 556.8°, and 63°, which can be indexed as (111), (311), (400), (422), (511), and (440); this is consistent with the standard card of Fe_3_O_4_ (JCPDS Card no. 19-0629). No diffraction peaks corresponding to other impurities were observed, indicating that the product was Fe_3_O_4_ with high purity. The FTIR spectra (Figure 2C) showed sharp peaks at 580, 1047, and 1629 cm^−1^, representing characteristic absorption peaks of the Fe–O bond, the stretching vibration of the CH_2_ bond, and the bending vibration of the NH bond, respectively [29]. This result indicated that the Fe_3_O_4_ surface was successfully modified with amino groups. The magnetisation curves revealed that the saturation magnetisation of Fe_3_O_4_ increased with the applied magnetic field strength and rapidly reached a saturated state (Figure 2D). The magnetisation curve appeared as an “S” shape (Figure 2D), indicating that the synthesised Fe_3_O_4_ exhibits superparamagnetism at room temperature.

### 3.3. UCNP Characterisation

To increase the solubility and stability of UCNPs in the water phase and to provide more modifiable active groups on the surface of the UCNPs, TEOS was used to form SiO_2_ on the surface, and the compound was modified with amino groups using APTES. The TEM images of the UCNPs showed that the particles were hexagonal in shape and approximately 100 nm in size (Figure 3A). After the UCNPs were coated with SiO_2_, a shell formed on their surface, which was clearly visible in the TEM images (Figure 3B). Figure 3C indicates that the UCNPs exhibited typical X-ray diffraction lines at 2θ = 29.95°, 30.79°, 34.69°, 39.61°, 43.42°, 46.84°, 53.56°, and 53.59° corresponding to the (110), (101), (200), (111), (210), (102), and (300) planes of NaYF_4_. According to the FTIR spectra, the stretching vibration peak of the Si-O bond at 1097 cm^−1^ and the bending vibration peak of the amino group at 1631 cm^−1^ appeared in the UCNPs@SiO_2_ and were absent in the untreated UCNPs (Figure 3D). This further supports the successful surface modification of the UCNPs. The fluorescence intensity of the two materials was measured using a fluorescence spectrophotometer with excitation and emission spectra of 980 nm and in the 400–800 nm region, respectively. The results showed that the fluorescence intensity of the modified UCNPs was much lower than that of the unmodified UCNPs, indicating that SiO_2_ affected their fluorescence intensity. Upon excitation with the 980 nm laser, the nanomaterials displayed three sets of fluorescence emission bands, of which the most obvious emission peak was at 542 nm (Figure 3E); therefore, this fluorescence value at 542 nm was chosen as the detection signal for the subsequent experiments.

### 3.4. Aptamer-Fe_3_O_4_ Magnetic Nanoparticles and Aptamer-UCNP Characterisation

The synthesised nanoparticles have a surface amino group, which can be cross-linked with the 5′-terminal-modified amino group aptamer using glutaraldehyde to enable functional modifications [30]. Since the aptamer is an oligonucleotide chain with a characteristic absorption peak at 260 nm [31,32], aptamer–nanoparticle binding was indirectly characterised by measuring the absorbance of the aptamer stock solution and the absorbance of the supernatant during binding. The UV-vis spectra results of the aptamer and aptamer-Fe_3_O_4_ magnetic nanoparticles (Figure 4A) show that the absorbance of the supernatant at 260 nm decreased significantly after the binding of different concentrations of aptamers, indicating that the aptamers successfully combined with the nanomaterials [33]. The capture capacity of aptamer-UCNPs was highly affected by the aptamers immobilised on the surface of the nanomaterials. Different reaction concentrations of UCNPs (1, 2, 4, and 6 mg/mL) were investigated to characterise the capture capacity and immobilisation efficiency of the aptamer-modified nanomaterials. The reduction in the absorbance of the supernatant at 260 nm increased as the concentration of UCNPs increased, indicating that more aptamer molecules were immobilised on the nanomaterial (Figure 4B). When the reaction concentration of the UCNPs was 6 mg/mL, the absorbance of the supernatant after the reaction was the lowest, indicating that a high initial reaction concentration could improve the immobilisation of the aptamer on the nanoparticles. Considering economy and efficiency, 6 mg/mL of UCNPs was selected for all subsequent experiments to synthesise the aptamer-UNCPs.

### 3.5. Optimal Amount of Aptamer-Fe_3_O_4_ Magnetic Nanoparticles and Aptamer-UCNPs

Different amounts of aptamer-Fe_3_O_4_ magnetic nanoparticles and aptamer-UCNPS were tested in the reaction system to optimise the detection method. Different amounts of Fe_3_O_4_ magnetic nanoparticles (40, 80, 120, 160, or 200 µL) were added to the reaction system. The fluorescence intensity gradually increased with the increased amount of added aptamer-Fe_3_O_4_ magnetic nanoparticles and peaked at 120 µL of aptamer-Fe_3_O_4_ magnetic nanoparticles (Figure 5A,B). Furthermore, the intensity decreased when more aptamer-Fe_3_O_4_ magnetic nanoparticles were added (Figure 5B). Thus, 120 µL of the aptamer-Fe_3_O_4_ magnetic nanoparticles was used for subsequent experiments. Similarly, when different amounts of aptamer-UCNPs (50, 100, 150, 200, or 250 µL) were added to the reaction system, the fluorescence intensity of the reaction system gradually increased as the amount of aptamer-UCNPs increased (Figure 5C). When more than 200 µL of aptamer-UCNPs was added, the fluorescence intensity of the system no longer increased (Figure 5D), indicating the aptamer-UCNPs were exceeded. Thus, 200 µL of the aptamer-UCNPs was used for the subsequent experiments.

### 3.6. Detection Method Sensitivity

The developed method was used to detect different concentrations of *V. parahaemolyticus* (with 120 µL of aptamer-Fe_3_O_4_ magnetic nanoparticles and 200 µL of aptamer-UCNPs used in the reaction system). The fluorescence intensities of the supernatants were measured at the end of the reaction. The results showed that the fluorescence intensity increased as the concentration of *V. parahaemolyticus* increased (Figure 6A) in the concentration range of 3.2 × 10^2^–3.2 × 10^5^ CFU/mL (Figure 6B). The linear equation was y = 296.40x − 217.67 (where x is log CFU mL^−1^ and y is the fluorescence intensity), with R^2^ = 0.9610, indicating that there was a positive correlation between the *V. parahaemolyticus* concentration and fluorescence intensity. The detection limit was 4.4 CFU/mL (3 ε/S, where ε is the standard deviation of the blank sample and S is the slope of the linear equation). Yu et al. reported a universal pathogen-sensing platform based on a smart hydrogel aptasensor embedded with gold nanoclusters to detect live *V. paraheamolyticus* in water with a detection limit of 10 CFU/mL [34]. Zhai et al. have developed an immunomagnetic separation and quantum-dot-based immunofluorescence method to detect *V. paraheamolyticus* with a detection limit of 10^2^ CFU/mL [35]. Similarly, Ren et al. used a fluorescence resonance energy transfer (FRET)-based paper sensor to detect *V. parahaemolyticus*; the detection limit was 8.9 CFU/mL [36]. Our results demonstrate that the fluorescence method we developed, based on magnetic and upconversion nanoparticles combined with aptamers, has sufficient sensitivity to detect *V. parahaemolyticus*.

### 3.7. Detection Method Specificity

To evaluate the specificity of the developed method for *V. parahaemolyticus*, four foodborne pathogenic bacteria, including *E. coli*, *enterotoxigenic E. coli*, *S. aureus*, and *Salmonella*, were added to the reaction system, and the fluorescence intensity of the supernatant after magnetic separation was measured. The tested *E. coli*, *Enterotoxigenic E. coli*, *S. aureus*, and *Salmonella* concentrations were 10^5^ CFU/mL; 10^5^ CFU/mL of *V. parahaemolyticus* was used as a positive control. The results revealed that only the separated *V. parahaemolyticus* suspension generated high fluorescence intensities, whereas the *E. coli*, *enterotoxigenic E. coli*, *S. aureus*, and *Salmonella* suspensions generated low fluorescence intensities using the developed method, proving that the detection method was highly specific for *V. parahaemolyticus* (Figure 7). Recently, Li et al. presented a colorimetric and SERS dual-mode detection method targeting *V. parahaemolyticus* using aptamers and multifunctional composite magnetic material, which exhibited no cross-reactivity with *E. coli*, *S. typhimurium*, *E. coli*, *S. aureus*, or *L. monocytogenes* [37]. Similarly, Parsaeimehr and Ozbay reported on a colorimetric detection method based on PCR and the utilisation of DNAzyme; their method also showed no cross-reactivity with *E. coli* or *S. aureus* [38]. Based on the results of the current study, the developed fluorescence method has good specificity for *V*. *parahaemolyticus*, is simple and sensitive, and can be adapted and applied to detect other foodborne pathogens.

## 4. Conclusions

In the present study, Fe_3_O_4_ magnetic nanoparticles and UCNPs were successfully synthesised. The Fe_3_O_4_ magnetic nanoparticles have a uniform size of 50 nm and exhibit superparamagnetism at room temperature. The TEM images of the UCNPs showed that the particles were hexagonal in shape and approximately 100 nm in size. The SiO_2_ synthesised UCNPs had three fluorescence emission bands, of which the most obvious emission peak was at 542 nm. After the immobilisation of the aptamer against *V. parahaemolyticus*, aptamer-Fe_3_O_4_ magnetic nanoparticles and aptamer-UCNPs were used as a capture medium and fluorescence signal source in detection, respectively. A fluorescence method for *V. parahaemolyticus* was designed using the compatibility of the aptamer and the target *V. parahaemolyticus*, which can rapidly isolate and detect *V. parahaemolyticus* from a complex matrix using magnetic forces. The developed fluorescence method-based aptamer-modified magnetic and upconversion nanoparticles had high specificity for *V. parahaemolyticus* and had no cross-reactivity with tested foodborne pathogenic bacteria, including *E. coli*, *enterotoxigenic E. coli*, *S. aureus*, and *Salmonella*. This method had a detection limit of 4.4 CFU/mL, the fluorescence signal had a clear linear relationship with the concentration of *V. parahaemolyticus* in the range of 3.2 × 10^2^–3.2 × 10^5^ CFU/mL, with a linear equation of y = 296.40x − 217.67 and a correlation coefficient of R^2^ = 0.9610. These results indicate that the developed fluorescence method can achieve the qualitative detection of *V. parahaemolyticus* in a complex matrix. Furthermore, this method can be adapted to detect other bacteria by changing the aptamers used and has broad application prospects in the field of pathogenic bacterial detection.

## Figures and Tables

**Figure 1 foods-12-04433-f001:**
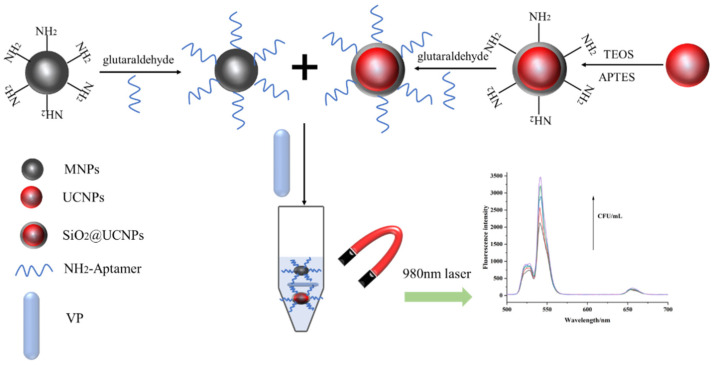
A schematic description of the proposed fluorescence detection platform.

**Figure 2 foods-12-04433-f002:**
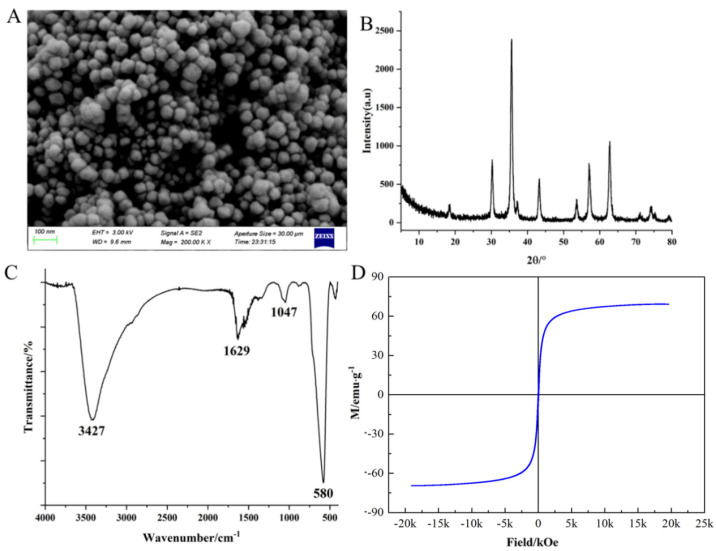
Characterisation of Fe_3_O_4_ magnetic nanoparticles modified with amino groups (TEM image of MNPs (**A**), XRD patterns of MNPs (**B**), FTIR spectra of MNPs (**C**), and VSM magnetisation curves of MNPs (**D**)).

**Figure 3 foods-12-04433-f003:**
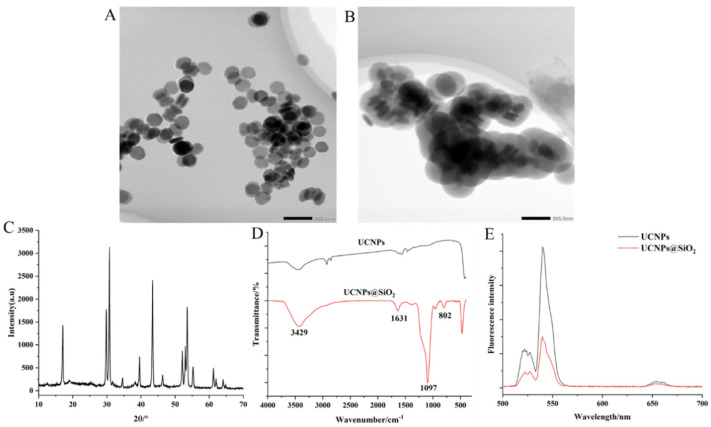
Characterisation of UCNPs and UCNPs@SiO_2_ (TEM image of UCNPs (**A**), TEM image of UCNPs@SiO_2_ (**B**), XRD patterns of UCNPs (**C**), FTIR spectra of UCNPs and UCNPs@SiO_2_ (**D**), and fluorescence spectrum of UCNPs, and UCNPs@SiO_2_ (**E**)).

**Figure 4 foods-12-04433-f004:**
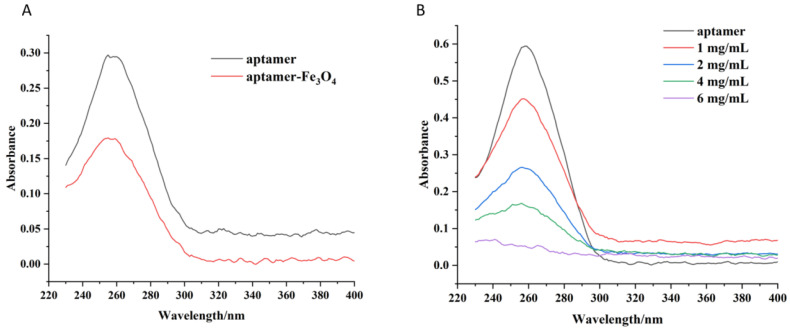
Confirmation of aptamer immobilisation on the Fe_3_O_4_ magnetic nanoparticles (**A**) and UCNPs (**B**) using UV-vis spectra.

**Figure 5 foods-12-04433-f005:**
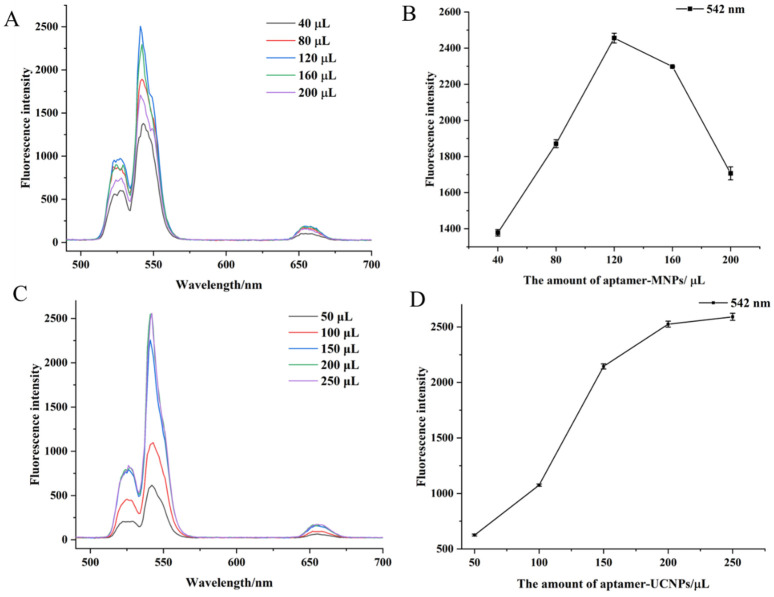
The fluorescence spectra of different additions of aptamer-MNPs (**A**), absorption peaks at 542 nm of different additions of aptamer-MNPs (**B**), the fluorescence spectra of different additions of aptamer-UCNPs (**C**), absorption peaks at 542 nm of different additions of aptamer-UCNPs (**D**).

**Figure 6 foods-12-04433-f006:**
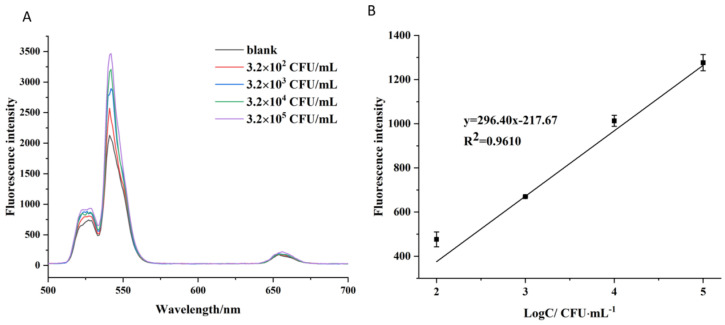
Fluorescence spectra of compounds in the presence of different *V. parahaemolyticus* concentrations (**A**), the linear relationship between the fluorescence intensity and concentration of *V. parahaemolyticus* (**B**), where F is the sample fluorescence intensity and F_0_ is the blank control.

**Figure 7 foods-12-04433-f007:**
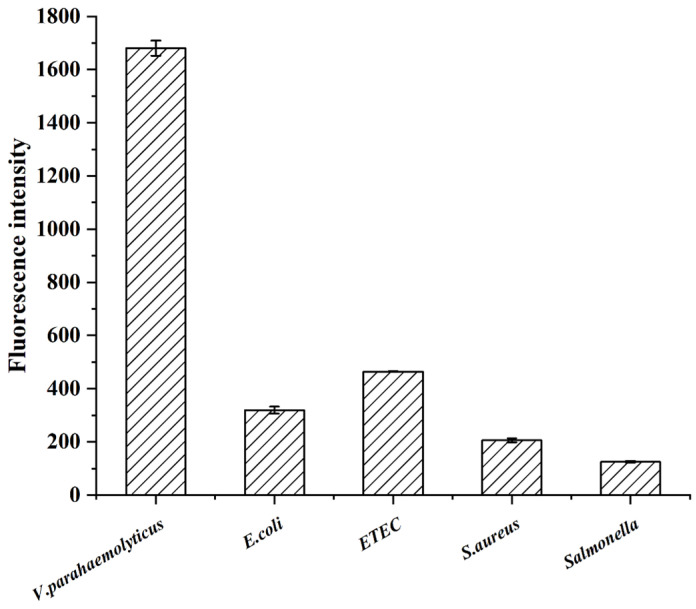
The specificity of the developed method with other bacteria.

## Data Availability

Data is contained within the article.

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
