# Peer review of "Detection of Vibrio parahaemolyticus Based on Magnetic and Upconversion Nanoparticles Combined with Aptamers"

_foods, 2023, doi:10.3390/foods12244433_

Round 1

Reviewer 1 Report

Comments and Suggestions for Authors

Comments and suggestions are in the attached file. 

Comments on the Quality of English Language

Quality of English language must be improved. 

Author Response

Thank you very much for taking the time to review this manuscript. Please find the detailed responses below and the corresponding revisions/corrections highlighted with red color in the re-submitted files.

Questions and Answers:

  1. Authors compare and discuss the luminescence intensity of UNCP nanoparticles and those after the silica coating. It is unclear from the represented results in what phase state are the both nanoparticles, is it aqueous phase, where uncoated UNCP tend to aggregate, thus producing lower luminescent signal that the colloidal stable silica coated UNCP. If the spectra are recorded in the dried state such comparison is meaningless, and quantum yields should be measured and applied for the compar

Answer to the reviewer:

Thank you so much for your valuable evaluation. In our study, both nanoparticles suspension were analyzed directly after uniformly dispersed in water phase by ultrasonic treatment to prevent the influence of aggregation  to the results. Related interpretation were added in MS Line 129-132 and highlighted with red color.

  1. It is incorrect to state out that the uncoated UNCP have no“surface modification groups”. Indeed, both OH- and F- should form an exterior layer of UNCP, although this layer in insufficient for their colloid stabi

Answer to the reviewer:

Thank you so much for your valuable evaluation. The incorrect statement was deleted, and the correction was added in MS Line 231-232 and highlighted with red color.

  1. The use ofglutaraldehyde as the linker for peptides should be followed by the cross-linking of the nanoparticles. This must be shown through the analysis of TEM images of both aptamer-MNPs and aptamer-UNCP

Answer to the reviewer:

Thank you so much for your valuable evaluation. It is true that TEM images is more intuitive to show the crosslinking of  aptamer and UNCPs. However, limited with our research condition, we use indirect method to evaluate the crosslinking of  aptamer and nanoparticles. We measured the aptamer in supernatant before and after reaction to confirm the  immobilization. Related contents are in line158-172 and  line 258-273.

  1. It is unclear (Fig. 5a) how the non-luminescent aptamer-MNPs can induce the luminescence. Maybe authors mean that MNPs affect the luminescence of UNCPs? Nevertheless, this subdivision must be rewritten.

Answer to the reviewer:

Thank you so much for your valuable evaluation. In this study, the aptamer-MNPs used as a capture agent of V. parahaemolyticus.  Aptamer-Fe3O4 magnetic nanoparticles could capture V. parahaemolyticus using aptamers when V. parahaemolyticus was present in the tested sample. The aptamer-Fe3O4-V. parahaemolyticus can be separated using a magnetic force. The separated aptamer-Fe3O4-V. parahaemolyticus was mixed with aptamer-UNCPs, a sandwich-like aggregates, aptamer-Fe3O4-V. parahaemolyticus-UNCPs can be formed and separated using a magnetic force. Due to the presence of UCNPs, the fluorescence intensity of the sandwich-like aggregate can be measured via fluorescence spectrophotometry with excitation spectra of 980 nm and emission spectra of 542 nm. Therefore, both aptamer-MNPs and aptamer-MNPs in reaction system will affect the luminescence. To find the optimal addition of aptamer-MNPs and aptamer-UNCPs, different amount of them were investigated. Results were presented in MS line 278-289.

  1. It is also unclear how the formation of the sandwich-like aggregates between UNCPs and MNPs has been detected. The formation of these aggregates is the key point in the  This issue must be clarified. 

Answer to the reviewer:

Thank you so much for your valuable evaluation. As we describe in Fig 1, both MNPs and UNCPs were modified with an aptamer which could specific identify and capture V. parahaemolyticus. Firstly, the tested sample was mixted with aptamer-MNPs and reacted under gentle shaking. Aptamer-MNPs could capture V. parahaemolyticusby aptamer when V. parahaemolyticus present in tested sample, and the aptamer-MNPs-V. parahaemolyticus could separate by a magnetic force. Next, the separated aptamer-MNPs-V. parahaemolyticus mixed with aptamer-UNCPs, a sandwich-like aggregates , aptamer-MNPs-V. parahaemolyticus-UNCPs can be formed and separated by by a magnetic force. Due to the presence of UCNPs, the fluorescence intensity of the sandwich-like aggregate can be measured via fluorescence spectrophotometry with an excitation spectra of 980 nm and emission spectra of 542 nm, and the intensity of this signal can be increased by increasing the concentration of V. parahaemolyticus for detection. Please see the related contents in MS Line 201-211.

Reviewer 2 Report

Comments and Suggestions for Authors

The paper “Detection of Vibrio parahaemolyticus based on magnetic and 2 upconversion nanoparticles combined with aptamers”, presented a well-argued description of a new method for diagnosis of the Vibrio parahaemolyticus.

The introduction was clear and exhaustive. The paragraph materials and methods described adequately all phases of study, but I suggest implementing the descriptions of the evaluation of the specificity of method with the use of the other bacterial strains. Overall the paper is very interesting for the scientific community and I propose it after minor revision.

Author Response

Thank you very much for taking the time to review this manuscript. More details on descriptions of the evaluation of the specificity of method with the use of the other bacterial strains were made according to reviewers comments and the changes were highlight with red color. Please see the change in text line 187-194. 

Reviewer 3 Report

Comments and Suggestions for Authors

The main objective of the study presented here is to develop a rapid and sensitive method for the detection of the foodborne pathogenic bacterium Vibrio parahaemolyticus. The proposed method uses aptamer-modified magnetic nanoparticles and up-conversion nanoparticles as capture and signal probes, respectively. To determine the specificity and sensitivity of this method, tests were carried out with different concentrations of V. parahaemolyticus and other foodborne pathogens such as E. coli, S. aureus and Salmonella.

In general, it can be said that the manuscript is well structured and generally well written. The methodology used is correct given the objective and is well supported by the bibliography. The presentation of the results is clear but should be improved. The interpretation of the results is done with the appropriate tools and methods and is also well supported by the literature.

Some minor corrections and suggestions are recommended:

11)     The phrase “using a fluorescence spectrophotometer with an external 980 nm laser” is repeated several times in the text (lines 153, 163, 172, 189 and 219) without any apparent reason. Actually, only the last sentence (line 219) is justified by the context. A revision of these passages is recommended.

22)     Line 259: there is a small typo in the unit superscript that should be corrected.

33)     Determination of specificity: The description of the methods used should be more detailed. In particular, the conditions used for each of these tests should be presented in more detail to avoid ambiguity and to improve the consistency of this report.

44)      There are some minor problems with the references that should be corrected: in reference #2 the link does not work, should be replaced by a valid reference; in reference #11 the DOI is not correct, it comes from another article; in reference #13 there is no DOI.

55)      In Figure 2, the material should not only be identified as MNP, in the XRD diagram (Fig. 2A) the peaks should be clearly identified with the corresponding orientation and source used; in the FTIR diagram (Fig. 2C) the peaks should also be identified in terms of their corresponding bonds; in the magnetisation diagram (Fig. 2D) two lines identifying both zero points of the axis should be added to facilitate the understanding of the data presented.

66)     Repeat in Figure 3 what was said for Figure 2 regarding the XRD and FTIR diagrams.

77)      Figure 4: The legend is unclear, and the corresponding line colours should be added to the legend.

The manuscript represents an original contribution and the study, and the results obtained are worth publishing after minor corrections.

Author Response

Thank you very much for taking the time to review this manuscript. Please find the detailed responses below and the corresponding revisions/corrections highlighted with red color in the re-submitted files

1) The phrase “using a fluorescence spectrophotometer with an external 980 nm laser” is repeated several times in the text (lines 153, 163, 172, 189 and 219) without any apparent reason. Actually, only the last sentence (line 219) is justified by the context. A revision of these passages is recommended.

Answer to the reviewer:

Thank you so much for your valuable evaluation. Correction have been made according to reviewers comments and highlighted with red color, please see the change in the text (lines 153, 163, 172, 189 and 219 →lines 166, 172, 193, 210 and 245)

2) Line 259: there is a small typo in the unit superscript that should be corrected.

Answer to the reviewer:

Thank you so much for your valuable evaluation. Correction have been made according to reviewers comments and highlighted with red color, please see the change in the text line 297.

3) Determination of specificity: The description of the methods used should be more detailed. In particular, the conditions used for each of these tests should be presented in more detail to avoid ambiguity and to improve the consistency of this report.

Answer to the reviewer:

Thank you so much for your valuable evaluation. Correction have been made according to reviewers comments and highlighted with red color, please see the change in the text line 312-330.

4) There are some minor problems with the references that should be corrected: in reference #2 the link does not work, should be replaced by a valid reference; in reference #11 the DOI is not correct, it comes from another article; in reference #13 there is no DOI.

Answer to the reviewer:

Thank you so much for your valuable evaluation. The references have been corrected according to reviewers’ comments and highlighted with red color.

5) In Figure 2, the material should not only be identified as MNP, in the XRD diagram (Fig. 2A) the peaks should be clearly identified with the corresponding orientation and source used; in the FTIR diagram (Fig. 2C) the peaks should also be identified in terms of their corresponding bonds; in the magnetisation diagram (Fig. 2D) two lines identifying both zero points of the axis should be added to facilitate the understanding of the data presented.

Answer to the reviewer:

Thank you so much for your valuable evaluation. Correction have been made according to reviewers comments and highlighted with red color, please see the change in the text line 216-224 for Fig 2A and Fig 2C. The Fig 2D also have corrected in Fig 2.

6) Repeat in Figure 3 what was said for Figure 2 regarding the XRD and FTIR diagrams.

Answer to the reviewer:

Thank you so much for your valuable evaluation. Correction have been made according to reviewers comments and highlighted with red color, please see the change in the text line 237-242.

7) Figure 4: The legend is unclear, and the corresponding line colours should be added to the legend.

Answer to the reviewer:

Thank you so much for your valuable evaluation. Correction have been made according to reviewers comments, please check the corrected Fig. 4.

The manuscript represents an original contribution and the study, and the results obtained are worth publishing after minor corrections.

Round 2

Reviewer 1 Report

Comments and Suggestions for Authors

I am satisfied by the revised version, thus, it diserves acceptance without further revision. 

Comments on the Quality of English Language

The work has been revised, and the quality of English

language is suitable for the acceptance.